# Marker-Assisted Backcrossing (MABc) to Improve Eating Quality with Thin Seed Coat and Aleurone Layer of Non-Glutinous Japonica Variety in Rice

**DOI:** 10.3390/genes13020210

**Published:** 2022-01-24

**Authors:** Me-Sun Kim, Ju-Kyung Yu, Seo-Rin Ko, Ki-Jo Kim, Hyeonso Ji, Kwon-Kyoo Kang, Yong-Gu Cho

**Affiliations:** 1Department of Crop Science, College of Agriculture and Life & Environment Sciences, Chungbuk National University, Cheongju 28644, Korea; kimms0121@cbnu.ac.kr (M.-S.K.); seorinko@cbnu.ac.kr (S.-R.K.); kgj2502@gmail.com (K.-J.K.); 2Syngenta Crop Protection LLC, Seeds Research, 9 Davis Dr. Research Triangle Park, Durham, NC 27709, USA; yjk0830@hotmail.com; 3Department of Agricultural Biotechnology, National Institute of Agricultural Sciences, Rural Development Administration (RDA), Jeonju 54874, Korea; jhs77@korea.kr; 4Division of Horticultural Biotechnology, Hankyong National University, Anseong 17579, Korea; kykang@hknu.ac.kr

**Keywords:** brown rice, cooking and eating quality, aleurone layer, MAB_C_, SNP, KASP marker

## Abstract

Brown rice is composed of rice bran, pericarp, seed coat, and aleurone layers, and the rice bran layer contains a large number of substances useful for the human body, such as dietary fiber, α-tocopherol, α-tocotrienol, and vitamins. However, more than 90% of these substances are removed when polished, and white rice has the disadvantage of losing food-related ingredients, such as umami-related amino acids, when compared to the unpolished group. In this study, we tried to develop new breeding lines with a thinner seed coat and aleurone layer to provide high eating quality with softer chewing characteristics and processability in rice grain. We detected an SNP for foreground selection for the backcross population by comparing genome sequences between Samgwang and Seolgaeng and developed high eating quality brown rice breeding lines by applying marker-assisted backcrossing (MAB_C_) breeding programs to backcross populations between Samgwang and Seolgaeng using KASP markers. SNP markers for foreground selection were identified to improve eating and processability through SNP mapping of Samgwang and Seolgaeng with SSIIa as a target gene in this study. Line selection according to genotype of KASP markers was successful in BC_1_F_1_ and BC_2_F_1_ generations, with the recurrent parent genome recovery ratio ranging from 91.22% to 98.65%. In BC_2_F_1_ seeds of the selected lines, thickness of the aleurone layer was found to range from 13.82 to 21.67 μm, which is much thinner than the 30.91 μm of the wild type, suggesting that selection by MABc could be used as an additional breeding material for the development of highly processed rice varieties. These lines will be useful to develop new brown rice varieties with softer chewing characteristics and processability in rice grain.

## 1. Introduction

Rice (*Oryza sativa* L.) is a vital, worldwide agricultural product. It is one of the leading food crops of the world, as more than half of the world’s population relies on rice as the major daily source of calories and protein [1]. Indeed, the demand for eating quality is continuously increasing around the globe because it is the most important factor in determining the market price [2]. The quality of rice used for eating and the rice yield potential is one of the main goals of rice breeding programs. The main components of rice grain quality include appearance, eating, cooking and milling quality, and nutritional qualities, which are all values that are determined by their physiochemical properties and other sociocultural factors [3].

Brown rice consists of 2–3% embryo, 92% endosperm, and 5–6% rice bran, which is composed of pericarp, seed coat, and an aleurone layer. The rice bran layer of brown rice contains dietary fiber and substances such as α-tocopherol, α-tocotrienol, γ-tocopherol, γ-aminobutyric acid (GABA), arabinoxylan, ferulic acid, and vitamins B1 and E. These contain physiologically functional substances such as antidiabetics, antihypertensives, and immune function enhancements, and are found in brown rice relatively more than other grains [4,5,6]. Brown rice is known to have a beneficial effect on human health because it is a whole-grain cereal. However, more than 90% of these substances are removed when polished, and white rice has the disadvantage of losing food-related ingredients, such as umami-related amino acids, when compared to the unpolished group [7]. Brown rice is rarely consumed as white rice is preferred for reasons related to appearance, taste, ease of preparation, tradition, safety, shelf life, and lack of awareness of benefits and availability, limiting its market potential [8,9,10].

‘Seolgaeng’ is a new *japonica* variety derived from ‘Ilpum’ mutant lines that uses treated, fertilized egg cells with N-methyl-N-nitrosourea (MNU) [11]. Seolgaeng is a soft rice with a round starch shape and is suitable as a processing rice, such as for brewing and for producing red yeast rice [12]. Seolgaeng is a non-glutinous rice with an amylose content of 18% or more, but it is easily crushed. In addition, Seolgaeng is a variety with a low protein content and high free sugar and essential amino acid content; it is mainly used for brewing or for the production of red yeast (*Monascus purpureus*) rice and yellow koji mold rice (*Aspergillus oryzae*) [12,13,14].

Molecular breeding that involves marker-assisted selection (MAS) addresses the limitations of conventional breeding and allows the pyramiding of multiple valuable genes into a single cultivar [15,16]. Utilization of DNA markers in a marker-assisted backcrossing (MABc) program significantly increases selection efficiency. The MABc approach avoids complicated issues associated with transgenic technology and conventional breeding methods by developing an ideal genotype in a short time [17,18,19]. A marker-assisted background selection can recover up to 99% of the recurrent parent genome in just three backcross cycles, whereas conventional breeding takes more than six backcrosses to recover 99% of the recurrent parent genome [20]. ‘KASP’ is a term that stands for Kompitive Allele Specific PCR, a novel competitive allelic PCR for SNP genotype analysis based on dual Fluorescent Resonance Energy Transfer (FRET). KASP is widely used for genetic mapping and trait-specific marker development such as rice genotype analysis, wheat leaf rust resistance analysis, and soybean cyst nematode resistance study; it is low-cost with high reliability and reproducibility [21,22,23].

In this study, we tried to develop new breeding lines with a thinner seed coat and aleurone layer to provide high eating quality with softer chewing characteristics and processability in rice grain by applying MABc breeding programs to backcross populations between Samgwang and Seolgaeng using KASP markers.

## 2. Materials and Methods

### 2.1. Plant Materials, Rice Cultivation, and Sample Preparation

In this study, rice varieties Samgwang [24] and Seolgang [11] were used as parents for developing a backcross population, and Koshihikari and Keunun were used as check varieties. They were cultivated and harvested according to a standard cultivation method of the Rural Development Administration (RDA) in Korea [25] in an experimental paddy field at Chungbuk National University in 2019–2021. Seeds were dried until a moisture content of 14% and we removed the hulls using a roller husking machine, and then brown rice was polished using a polishing machine (MC-90A, Toyoseiki, Tokyo, Japan) to a degree of grinding rate of 90%. Fifteen grams of polished rice were made of rice flour using a 100-mesh screen by Cyclotec Sample Mill (Tecator Co., Höganäs, Sweden), which was used for viscosity analysis.

### 2.2. MABc Breeding Strategy

Samgwang, an excellent variety cultivated in a wide area in Korea, was used as a female parent and crossed with Seolgaeng as a male parent to develop F_1_ generation in an experimental paddy field at Chungbuk National University in 2019. F_1_ plants were backcrossed to Samgwang for BC_1_F_1_ generation. Consecutive backcrossing was conducted until BC_2_F_1_ generation was produced from 2019 to 2020. KASP markers were applied to each generation for background selection. Out of a total of 210 individuals in BC_1_F_1_, 96 plants were selected through the foreground selection. The genetic background of those 96 individuals was screened, and six selected individuals were progressed to the BC_2_F_1_ population. The BC_2_F_1_ population was developed in the experimental paddy field, and their phenotype was observed through agronomic trait analysis. As in BC_2_F_1_, 15 individuals were selected through foreground and background selection in BC_2_F_2_. Then, 15 BC_2_F_2_ seeds were used to investigate characteristics related to eating quality.

### 2.3. Evaluation of Cooking and Pasting Characteristics

#### 2.3.1. Analysis of Amylose and Protein Contents, Whiteness

Amylose content and protein content and whiteness (an indicator of the whiteness of grains of rice) were measured using the Infratec 1241 Grain Analyzer (FOSS, Hilleroed, Denmark) and were investigated in three replicates for each treatment [26]. Near-infrared spectroscopy is a device that detects the content of components such as carbohydrates, protein, fat, and moisture by measuring absorbance using electromagnetic waves (780–2500 nm). This instrument has the advantage of being fast and non-destructive because it is simple to use and measures several components simultaneously [27].

#### 2.3.2. Characterizations of Cooking and Eating Texture

Thirty grams of milled rice samples were cooked for 30 min and cooled down for 25 min using a cooler. Ten grams of cooked milled rice were placed on an experimental plate and used for investigating cooking taste characteristics by a full cup method [28]. Each experiment was conducted in three replicates. A TensiPresser Analyzer (My Boy, TAKETOMO Electric Inc., Tokyo, Japan) was used to investigate cooking and eating quality and to measure hardness, adhesiveness, springiness, stickiness, and thickness.

#### 2.3.3. Rapid Visco Analysis (RVA)

Three grams of rice flour with 25 mL of distilled water were blended in an aluminum can and transferred to a Rapid Viscosity Analyzer (Model RVA-4, Newport Scientific Ltd., Warriewood, Australia). Analysis conditions for RVA were set to be started at 50 °C for one minute and heated to 95 °C at the rate of 12 °C per min. Then, the temperature was set to remain at 95 °C for two min and cool down to 50 °C for 7 min. Each experiment was conducted in three replicates. Six paste viscosity properties were derived through this process, denoted as PV (peak viscosity, RVU), HPV (hot paste viscosity, RVU), CPV (cool paste viscosity, RVU), Breakdown (highest viscosity-lowest viscosity, RVU), Setback (cool paste viscosity-highest viscosity, RVU), and GT (gelatinization temperature, °C).

### 2.4. Molecular Marker Analysis

#### 2.4.1. Foreground Selection Using Semi-Nested PCR Analysis

Whole genome re-sequencing analysis of Samgwang and Seolgaeng was performed using HiSeq 2500 Sequencing System (Illumina, San Diego, CA, USA), and short-read sequences obtained from the genome re-sequencing data were aligned through the Bowtie program. Assembly and mapping were performed using CLC Main Workbench Software (QIAGEN, Hilden, Germany). Mutation analysis was performed by comparison with reference rice genome sequence of IRSGP1.0 (International Rice Genome Sequencing Project). Foreground selection markers were designed at ranges up to 1000 bp, including the mutant region of the OsSSIIa gene. To select individuals with target genotypes, DNA was extracted from leaves of Samgwang/Seolgaeng BC_1_F_1_ and BC_2_F_1_ populations and deep-sequencing was performed based on semi-nested PCR using the method described by Chi et al. [29].

#### 2.4.2. Background Selection Using KASP Marker Analysis

In order to detect SNPs between Korean Japonica rice cultivars, 773 KASP markers were used to select lines with high recurrent parent genome recovery ratios for BC_1_F_1_ and BC_2_F_1_ populations between Samgwang and Seolgaeng [30]. For analysis using KASP markers, polymorphisms of Samgwang and Seolgaeng were investigated through Seed Industry Promotion Center of Foundation of Agri. Tech. Commercialization and Transfer (FACT) (Gimje, Korea). Among 386 KASP markers showing polymorphism, markers showing hetero genotypes were excluded, and 96 markers located at 5 Mb intervals per 12 chromosomes in rice were selected to analyze the recurrent parent genome recovery ratios in BC_1_F_1_ and BC_2_F_1_ of Samgwang/Seolgaeng. A genetic graphic map was derived using a MapChart program (version 2.32) based on the physical locations of the KASP markers on chromosomes [31].

#### 2.4.3. Investigation of Agronomic Traits

Agronomic characteristics such as plant height, culm length, panicle length, and number of tillers in the lines selected from the BC_1_F_1_, BC_2_F_1_, and BC_2_F_2_ generations were investigated for phenotypes according to a standard method of RDA, Korea [25]. Using the 15 BC_2_F_2_ lines that show a recovery rate of 97–99.1% of the recurrent parent genome, paste viscosity characteristics were investigated as an eating quality-related trait. Each experiment was conducted in three replicates.

### 2.5. Histological Study

Histological analysis of seed coat and aleurone layer structures were performed according to Carlo and Marco [32]. Immature seeds were collected 18 days after flowering on plants. Each experiment was conducted in three replicates. Collected samples were fixed in FAA (formalin–acetic acid–ethanol) for 48 h. Then, samples were dehydrated in a graded series of ethanol (60%, 70%, 80%, 90%, 95%, and 100%) for 30 min each and embedded in paraffin wax. Embedded samples were cut into 20 μm sections and were stained using 0.5% periodic acid and Schiff reagent. After staining, seed coat and aleurone layers structures were examined using an Eclipce E600 microscope (Nikon, Tokyo, Japan).

### 2.6. Statistics Analysis

SAS program (SAS Institute Inc., Cary, NC, USA) was used for statistical analysis of the investigated cooking and eating quality traits. Basic statistics such as mean and standard deviation were analyzed according to the collected data, and the distribution of variance was investigated. Significance (*p* < 0.05) was analyzed using analysis of variance, and significant differences were compared and analyzed by performing Duncan’s Multiple Range Test.

## 3. Results

### 3.1. Evaluation of Characteristics Related to Eating and Cooking Quality in Parents

Samgwang is a variety with high yield and high quality and is one of the most popular in Korea [33]. Seolgaeng is an excellent rice variety that consumers use as a brown rice. Compared to white rice, brown rice is rich in nutrients such as protein, dietary fiber, and vitamins, so it is highly preferred by consumers as a health functional food. However, the rice bran layer is thicker than that of white rice and the texture of the rice is somewhat rough because moisture absorption is poor due to the difference in structural characteristics [11,12]. Prior to using Samgwang and Seolgang as experimental materials, cooking and texture characteristics, amylose content, protein content, whiteness, and viscosity characteristics were investigated to compare eating quality-related characteristics of Samgwang and Seolgaeng. At the same time, considering that Samgwang is a white rice variety and Seolgaeng is a brown rice variety, they were compared with Koshihikari and Keunnun, respectively.

#### 3.1.1. Characteristics Related to Cooking and Texture

Table 1 shows the cooking and texture characteristics of Samgwang, Koshihikari, Seolgaeng, and Keunnun. Hardness is the force required to compress the rice when chewing, and the higher it is, the harder the texture is evaluated. The hardness levels of Samgwang and Koshihikari were measured at 36.03 and 31.75, compared to those of Seolgaeng and Keunnun, which were 38.43 and 37.33, respectively. The white rice variety is expected to have a lower hardness than the brown rice variety, causing it to be softer to chew. Adhesiveness is the tenacity of the rice. Koshihikari (71.92) had higher adhesiveness than Samgwang (64.69), and Seolgaeng (53.02) had higher adhesiveness than Keunnun (52.09). Springiness refers to the degree to which a grain of rice returns when pressed but not completely broken. Koshihikari (34.56) was higher in springiness than Samgwang (30.92), and Seolgaeng (37.48) was the highest among the four varieties. As for stickiness, Koshihikari (75.10) was measured as higher than Samgwang (68.73), and Seolgaeng was 56.89, which was higher than Keunnun (47.32).

#### 3.1.2. Characteristics Related to Eating Quality

Eating quality-related characteristics such as amylose content, protein content, and whiteness of Samgwang, Koshihikari, Seolgaeng, and Keunnun are shown in Figure 1. Amylose content of all four varieties was within the range of amylose content in non-glutinous rice for cooking (18–20%), but it was found that the amylose content was rather high at 19.3% in Seolgaeng. It is known that the protein content of rice is inversely proportional to the taste of rice during cooking and proportional to its viscosity. In addition, the higher the protein content, the more transparent and harder the rice grains are, so it takes more water and time to cook [34]. Protein content of the four varieties was similar to that of *japonica* rice at 5.91–7.89% [35], and there was no significant difference between the varieties. Whiteness is an indicator of the phenotype of rice grain, with Seolgaeng showing the highest value at 55%, followed by Koshihikari at 44.6%, Keunnun at 43.1%, and Samgwang at 31.5%.

#### 3.1.3. Viscosity Characteristics by RVA

When starch is heated above gelatinization temperature with sufficient water, starch particles absorb water and swell, thereby increasing the volume and viscosity [36]. Gelatinization initiation temperature is the minimum temperature required for gelatinization and the temperature at which the viscosity starts to increase. Varieties with a high gelatinization temperature require a lot of water and time to cook, and consumers prefer varieties with a lower temperature [37]. When comparing the gelatinization temperatures of the four varieties, Samgwang, Koshihikari, and Keunnun, having similar amylose content showed similar gelatinization temperatures, and Seolgaeng showed a higher gelatinization temperature (Table 2). In general, rice with a high amylose content has a higher gelatinization initiation temperature because the starch structure is denser [33,38]. High viscosity indicates the water capacity of starch, and the higher the peak viscosity, the weaker the gel is that is forms. Samgwang showed a higher viscosity than other varieties, and it is estimated that it will be softer when gelatinized. If the starch is continuously heated after reaching the highest viscosity, the amylose that was forming the starch particles is dissolved and the starch particles break, thereby reaching the lowest viscosity at which viscosity decreases. Breakdown viscosity, which is the difference between peak viscosity and hot paste viscosity, is related to the stability of processing, and the lower the value, the stronger the resistance to heat and shear of gel, which is considered favorable for processing [39]. The breakdown viscosity was shown in order of Samgwang, Koshihikari, Seolgaeng, and Keunnun, indicating that Seolgaeng and Keunnun are the most suitable for processing. This is also consistent with the report that Seolgaeng and Keunnun are used for brewing and for the production of germinated brown rice [40,41]. When the gelatinized starch is cooled, the eluted amylose molecules are entangled to form a network structure, and the empty space is also filled with broken starch particles, thereby increasing viscosity as the structure hardens. The maximum recorded viscosity is called final viscosity or cool paste viscosity. The higher the amylose content, the faster the structure is formed, so the cool paste viscosity increases [42]. Samgwang and Koshihikari showed similar cool paste viscosity, and the processing varieties Seolgaeng and Keunnun showed similar cool paste viscosity. Although the highest amylose content was in Seolgaeng, Keunnun showed the highest cool paste viscosity, suggesting that aging occurs long after cooking. Setback viscosity is the value obtained by subtracting the peak viscosity from the cool paste viscosity, and the higher the value of setback viscosity, the faster the starch aging occurs [39]. The order of setback viscosity from highest to lowest was Keunnun, Seolgaeng, Koshihikari, and Samgwang; therefore, it is thought that Samgwang has the slowest aging and Keunnun has the fastest aging.

### 3.2. Whole-Genome Re-Sequencing Analysis and Variant Discovery for Foreground Selection

#### 3.2.1. Foreground Selection in BC_1_F_1_ and BC_2_F_1_

Using SNP found in the SSIIa gene region between Samgwang and Seolgaeng, NGS analysis was performed to select lines with the SSIIa genotype of Seolgaeng from the BC_1_F_1_ and BC_2_F_1_ populations. The structure of the SSIIa gene showing the mutation between Samgwang and Seolgaeng and the location and amino acid mutation of the discovered SNP are shown in Figure 2A. Two PCRs were performed to confirm the target SNP. The PCR test was performed by synthesizing the first primer having a size of 782 bp, including the target SNP position in the SSIIa gene. Then, PCR was performed on the first PCR product by designing a second primer having a size of 276 bp (Figure 2B). After confirming the presence of the PCR band by electrophoresis, NGS analysis was performed on BC_1_F_1_ (210) and BC_2_F_1_ (96) individuals. As a result, it was confirmed that 100 individuals in BC_1_F_1_ and 55 individuals in BC_2_F_1_ had the SNP type of SSIIa as heterozygosity (G/A). Finally, it was confirmed that the separation ratio was 1:1 in BC_1_F_1_ and BC_2_F_1_ generations. Therefore, background selection analysis was performed on these selected individuals (Table 3).

#### 3.2.2. Background Selection in BC_1_F_1_ and BC_2_F_1_

Prior to the selection of individuals with a high recurrent parent genome recovery ratio in the BC_1_F_1_ and BC_2_F_1_ populations of Samgwang and Seolgang, the KASP marker validity test as a background selection marker for Samgwang and Seolgang was performed. Among 773 KASP markers developed to detect SNPs between japonica rice varieties, a total of 397 KASP markers were found to distinguish genotypes of Samgwang and Seolgaeng, excluding markers that were not amplified or that showed heterozygote genotypes. Among these, the final 96 KASP markers were selected at about 5 Mb intervals for each chromosome to be evenly distributed in 12 rice chromosomes (Figure 3). Using the final selected 96 KASP markers, genotyping analysis was performed on 96 BC_1_F_1_ (Figure 4A) and 55 BC_2_F_1_ (Figure 5A) individuals from Samgwang and Seolgaeng chosen in foreground selection, respectively. As a result of analyzing the recovery ratio of the recurrent parent genome of 96 individuals of Samgwang and Seolgaeng BC_1_F_1_ and based on the genotyping results, most of them showed a range of 60% to 65%, and seven individuals showed a recovery ratio of 75% or more. Among the individuals with a recurrent parent genome recovery ratio in the range of 75% or more, seven BC_1_F_1_ lines with excellent phenotypes were selected (Figure 4B). Table 4 shows the results of the investigation of plant height, culm length, panicle length, and number of tillers. SS007 showed the highest number of tillers, SS045 showed the longest length, and SS100 was confirmed to have the longest plant length. The selected BC_1_F_1_ individuals were crossed with Samgwang to produce the next BC_2_F_1_ backcross generation (Figure 5B). As a result of agriculture traits analysis of BC_2_F_1_ individuals of Samgwang and Seolgaeng, all six BC_2_F_1_ individuals exhibited shorter panicle length than those of Samgwang, but plant height and culm length showed similar levels to Samgwang. Therefore, it is estimated that individuals with a high recurrent parent genome recovery ratio will exhibit excellent yield due to having a large number of tillers while showing a phenotype similar to Samgwang (Table 5).

### 3.3. Viscosity Properties of the Selected Lines in BC_2_F_2_

Characteristics of paste viscosity were investigated for 15 lines, including the selected lines, whose target SNP of the SSIIa gene had a Seolgaeng genotype and whose genome recovery ratio of Samgwang was 98% or higher (Table 6). Gelatinization initiation temperature of 15 lines showed a similar trend overall. Peak viscosity of nine lines (SS20-04, SS29-25, SS29-26, SS29-11, SS29-12, SS50-25, SS50-26, SS50-27, and SS50-28) was higher than Samgwang, and breakdown viscosity similar to Samgwang was found in eight lines. Cool paste viscosity was found between Samgwang and Seolgaeng, and the lowest viscosity of setback viscosity was found in SS29-26 and the highest in SS50-03. Nine BC_2_F_2_ lines, which have higher peak viscosity than Samgwang, have a high cool paste viscosity, meaning they are expected to form a soft gel during gelatinization. In addition, all 15 lines are expected to show cool paste viscosity similar to Samgwang, which is expected to cause rapid aging of starch; however, some lines with higher setback viscosity than Samgwang, such as SS50-26, are expected to show lower setback viscosity, resulting in higher processing stability and slower aging of starch.

### 3.4. Histochemical Analysis of Seed Coat and Aleurone Layers in BC_2_F_1_

For the histological analysis of seed coat and aleurone layer thickness of Samgwang and Seolgaeng, 18 DAF seeds were cut with a microtome, and tissue staining by carbohydrate staining was performed (Figure 6). As a result, it was confirmed that the thickness of the aleurone layer in Samgwang was 30.91 μm, which was thicker than the aleurone layer in Seolgaeng that was measured at 11.93 μm. As a result of performing histological analysis using a microtome on four BC_2_F_1_ seeds with the highest recurrent parent genome recovery ratio, it was confirmed that they had similar seed coat and aleurone layer thickness to Seolgaeng, with high eating quality and processable brown rice variety characteristics (Figure 7). Therefore, four BC_2_F_1_ lines (SS50-25, SS50-26, SS50-27, and SS50-28) with a high recurrent parent genome recovery ratio and similar seed coat and aleurone layer thicknesses to Seolgaeng among BC_2_F_1_ cultivated in this study can be used as excellent breeding materials with improved eating quality and processability.

### 3.5. Agronomic Traits of the Selected Lines in BC_2_F_2_

The selected and improved BC_2_F_2_ lines were evaluated based on agriculture characteristics and the results were compared to the phenotype of their recurrent parent, Samgwang (Figure 8). Plant height ranged from 104.8 cm to 112.8 cm, which was similar to the plant height of Samgwang, the recurrent parent variety. Culm length was 85.0 cm to 94.5 cm, similar to the 88.7 cm of Samgwang. The lines were also similar to Samgwang in panicle length and number of tillers (Table 7). Therefore, it was confirmed that BC_2_F_2_ lines with high recurrent parent genome recovery ratios showed a similar phenotype to Samgwang.

## 4. Discussion

Rice analysis using an RVA profile can characterize changes in physicochemical properties of rice during starch kneading and show a significant correlation with meal quality [43,44]. The RVA profile is closely related to the amylose content and diet in rice [45,46]. The elements of an RVA profile such as breakdown viscosity and setback viscosity can reflect the cooking and eating quality differences between rice varieties [47], and can also be used as an auxiliary means to evaluate the quality of rice intake [48,49]. Wu et al. [50] revealed that the soluble starch synthase gene *SSIIa* has a significant influence on the RVA profile properties of glutinous rice. Previous studies have shown that the cooking and eating quality of rice is regulated by other genes related to starch synthesis in addition to the *Wx* gene. Tian et al. [44] revealed that *SSIIa* affects amylose content, gel consistency, and gelatinization temperature, and *SSI*, *SSIIIa*, *AGPase*, and *PUL* also affect amylose content by association analysis.

The degree of cooking of rice grain with different degrees of milling showed that the aleurone layer affects the degree of rice cooking, including factors such as optimum cooking time and water absorption, as well as affecting texture profile features such as hardness, adhesiveness, paste profile, and characteristics of viscosity [51,52,53]. However, rice grain has an uneven surface, making it difficult to remove the aleurone layer by mechanical milling [54,55]. *SS2*, encoded by the Sus-1 gene in maize, is mainly expressed in the aleurone layers and the embryo. Some *SS2* proteins are also detected in endosperm tissues, but are mainly localized at low levels in the crown region and basal transfer cells [56,57]. Similarly, in situ hybridization experiments show that *SS1* RNA mainly accumulates in the endosperm tissues except for the aleurone layer, whereas *SS2* RNA accumulates in the aleurone layer and the embryo [58].

Therefore, SNP markers for foreground selection were identified to improve edibility and processability through SNP mapping of Samgwang and Seolgaeng with SSIIa as a target gene in this study. Line selection according to the genotype of the KASP marker was successful in BC_1_F_1_ and BC_2_F_1_ generations with a recurrent parent genome recovery ratio ranging from 91.22% to 98.65%. We applied SNP markers to shorten the breeding cycle for background selection with the KASP markers. SNPs for KASP markers related to cooking and eating quality were discovered through genome mapping after whole genome re-sequencing analysis. The conventional backcrossing process takes six to seven generations or more, but the MABc system can shorten the breeding cycle to three to four generations or fewer, thereby reducing the number of breeding cycles [59]. It has been successfully applied to various crops such as rice, soybean, and rye, and has successfully contributed to the development of new varieties [22]. Currently, KASP analysis is being used in various crops due to its low cost and locus specificity and efficiency [60].

In this study, 397 polymorphic markers were identified out of 773 KASP markers tested in both parents. As a result of final analysis with 397 KASP markers for background selection analysis, 96 KASP markers were selected with an interval of about 5 Mb per chromosome and were evenly distributed across 12 rice chromosomes. The recurrent parent genome recovery ratio was analyzed using 96 markers. Twenty-five lines with recurrent parent genome recovery ratios in the range 91.22–98.65% were selected, and finally, six lines with excellent phenotypes in BC_2_F_1_ were selected. In our investigation of agricultural traits, most traits were identified to recover the traits of Samgwang well due to a high genomic recovery ratio through backcrossing of Samgwang and Seolgaeng. In BC_2_F_1_ seeds of the selected lines, thicknesses of the seed coat and aleurone layer were found to range from 13.82 to 21.67 μm, suggesting that selection by MABc could be used for developing softer chewing characteristics of rice varieties. Since this study performed MABc on BC_1_F_1_ and BC_2_F_1_ lines in the early generation of backcross breeding, the best performing rice lines were selected in a relatively short breeding cycle. We believe that the seed coat and aleurone layer thickness mutant lines for better eating quality and processability can be used as new breeding materials to develop softer and high-quality rice varieties in rice breeding programs.

## 5. Conclusions

Consumers prefer to have high eating quality rice with high nutrients. Brown rice is rich in nutrients such as protein, dietary fiber, and vitamins. However, the rice bran layer is thicker than that of white rice and the texture of the rice is somewhat rough because moisture absorption is poor due to the difference in structural characteristics. In this study, we aimed to breed new rice lines with a thinner seed coat and aleurone layer to provide high eating quality with softer chewing characteristics and processability in rice grain by applying marker-assisted backcrossing (MABc) breeding programs to backcross populations between Samgwang and Seolgaeng using KASP markers.

As a result, we selected six lines that were detected to range from 13.82 to 21.67 μm in thickness of the aleurone layer, which is much thinner than the 30.91 μm of the wild type. These lines will be useful to develop new brown rice varieties with softer chewing characteristics and processability in rice grain.

## Figures and Tables

**Figure 1 genes-13-00210-f001:**
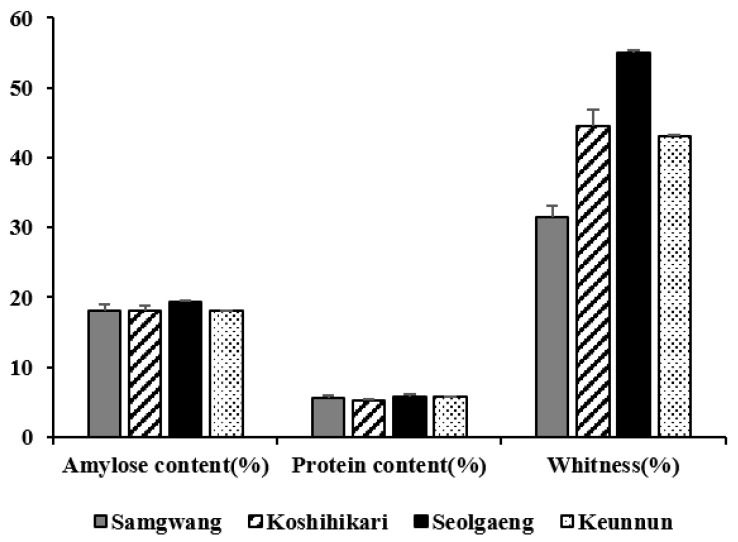
Comparison of amylose content, protein content, and whiteness, which are related to eating quality in four Korean rice varieties. Each experiment was performed in three replicates.

**Figure 2 genes-13-00210-f002:**
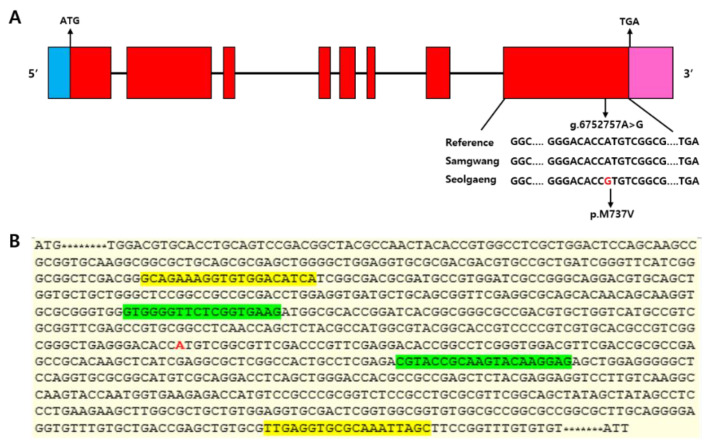
Structure of SSIIa gene contained SNP location and amino acid substitution. (**A**) SNP loci(A/G) showing missense non-synonymous amino acid substitution [Methionine(M) to Valine(V): p.M737V] in 8th exon of SSIIa gene. Blue box, upstream; red box, CDS; pink box, downstream. (**B**) Positions and information of 1st and 2nd PCR primer sets and SNP (red color) used for foreground selection in CDS region of SSIIa gene; 1st PCR primer set is yellow color, 2nd PCR primer set is green color.

**Figure 3 genes-13-00210-f003:**
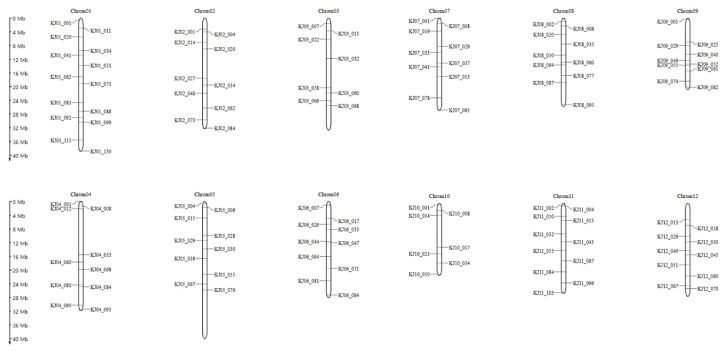
Chromosomal locations of 96 KASP markers indicating SNP polymorphism between Samgwang and Seolgaeng marked at approximately 5 Mb intervals on each chromosome for background selection.

**Figure 4 genes-13-00210-f004:**
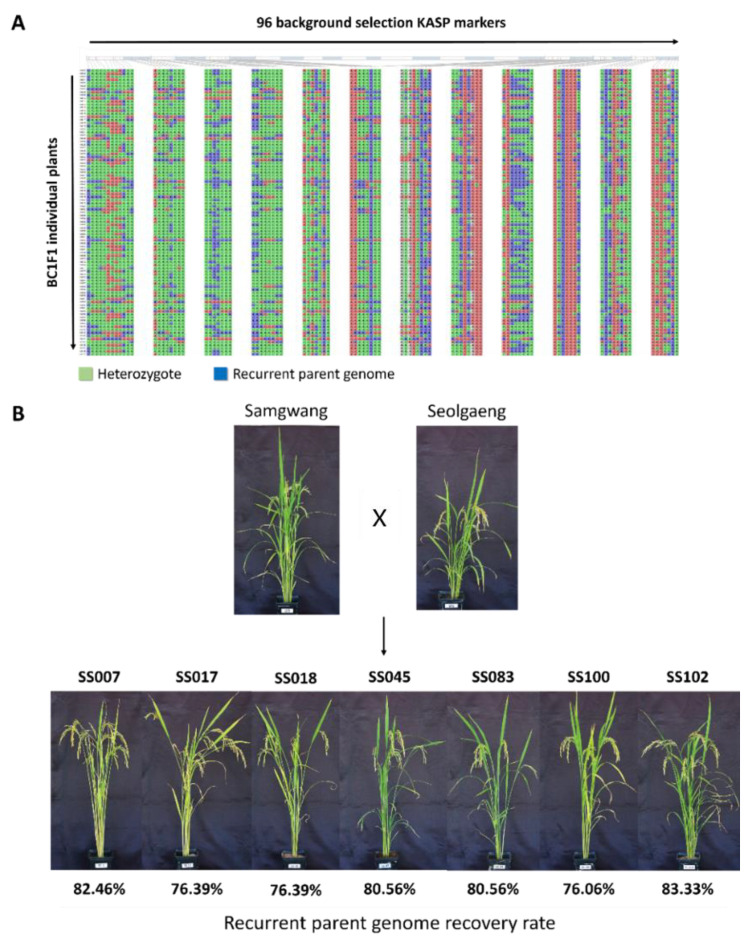
(**A**) Genotyping results using 96 KASP markers for the BC_1_F_1_ population of a total of 93 individuals. (**B**) Phenotype observation of seven BC_1_F_1_ lines with high genome recovery rates for recurrent parent, Samgwang.

**Figure 5 genes-13-00210-f005:**
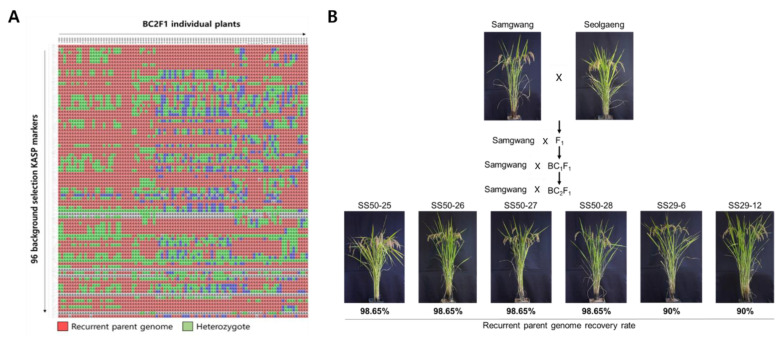
(**A**) Genotyping results of 93 BC_2_F_1_ plants using 96 KASP markers for background selection in the backcross population between Samgwang and Seolgaeng. (**B**) Phenotype observation of six BC_2_F_1_ lines with high genome recovery rates for recurrent parent, Samgwang.

**Figure 6 genes-13-00210-f006:**
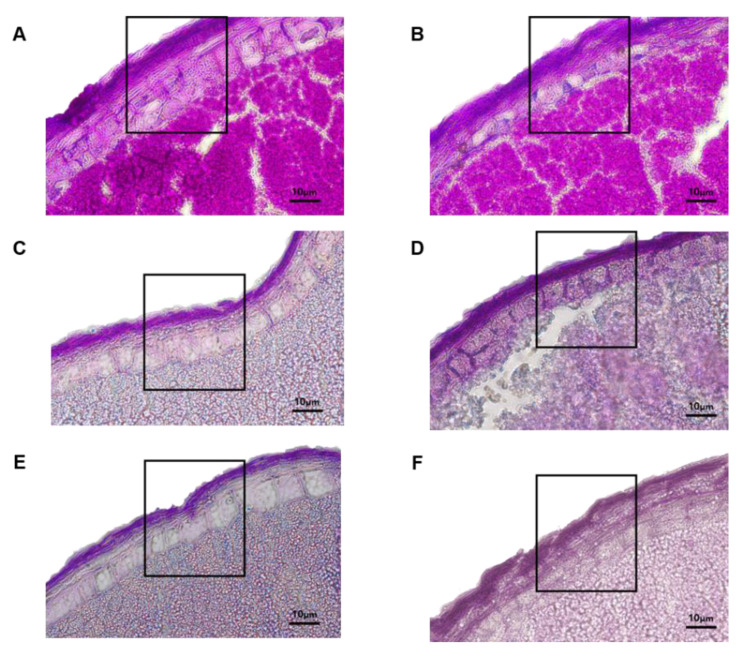
Histological observation of seed coat and aleurone layers in the selected BC_2_F_1_ grains compared with Samgwang and Seolgaeng. The square box indicates region of seed coat and aleurone layer investigated in this study. (**A**): Samgwang (30.91 μm), (**B**): Seolgaeng (11.83 μm), (**C**): SS50-25 (15.68 μm), (**D**): SS50-26 (15.42 μm), (**E**): SS50-27 (13.82 μm), (**F**): SS50-28 (21.67 μm). Size bars = 10 μm.

**Figure 7 genes-13-00210-f007:**
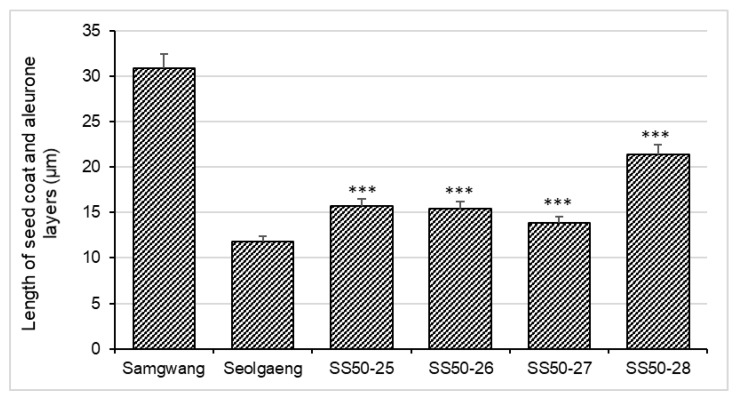
Length of seed coat and aleurone layers of selected lines in comparison with parents. Each experiment was performed in three replicates. ***: *p* < 0.001.

**Figure 8 genes-13-00210-f008:**
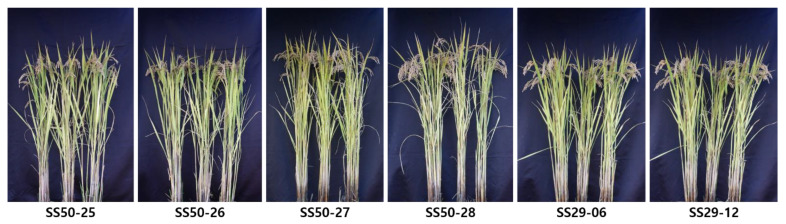
Observation of phenotypes of BC_2_F_2_ lines advanced in rice paddy field.

**Table 1 genes-13-00210-t001:** Characteristics of texture and eating quality in Samgwang, Koshihikari, Seolgaeng, and Keunnun.

Variety	Hardness	Adhesiveness	Springiness	Stickiness
Samgwang	36.03 ± 0.26 ^§^	64.69 ± 11.75	30.92 ± 0.94	68.73 ± 8.25
Koshihikari	31.75 ± 5.73	71.92 ± 14.79	34.56 ± 4.26	75.10 ± 9.91
Seolgaeng	38.43 ± 2.81	53.02 ± 6.84	37.48 ± 3.18	56.89 ± 8.68
Keunnun	37.33 ± 5.29	52.09 ± 8.59	23.62 ± 3.96	47.32 ± 13.38
*F*-value	1.494	2.301	9.577 **	4.357 *

* *p* < 0.05; ** *p* < 0.01. ^§^ Each test was performed in three replicates and values are expressed as average ± standard deviation.

**Table 2 genes-13-00210-t002:** Pasting properties in Samgwang, Koshihikari, Seolgaeng, and Keunnun.

Variety	Viscosity (RVU)					GT ^∫^
	PV ^§^	HPV ^¶^	Breakdown	CPV ^†^	Setback	(°C)
Samgwang	296.42 ± 4.92 ^∫^	170.86 ± 6.74	125.56 ± 18.86	253.21 ± 9.33	−43.21 ± 13.73	69.73 ± 0.45
Koshihikari	264.38 ± 9.32	160.70 ± 15.18	103.68 ± 11.46	252.03 ± 11.71	−12.35 ± 9.44	70.67 ± 1.19
Seolgaeng	251.89 ± 5.01	159.94 ± 2.76	91.95 ± 2.46	260.22 ± 3.43	8.33 ± 1.92	72.13 ± 0.43
Keunnun	226.42 ± 2.06	154.61 ± 4.67	71.81 ± 3.56	263.92 ± 3.85	37.50 ± 1.91	70.92 ± 0.36
*F*-value	14.253 **	6.997 *	2.988	6.449 *	14.647 *	55.057 ***

^§^ PV, peak viscosity; ^¶^ HPV, hot paste viscosity; ^†^ CPV, cool paste viscosity; ^∫^ GT, gelatinization temperature. * *p* < 0.05; ** *p* < 0.01; *** *p* < 0.001. *^∫^*Each test was performed in three replicates and values are expressed as average ± standard deviation.

**Table 3 genes-13-00210-t003:** Statistical analysis for genotyping results through foreground selection with 210 backcross population (BC_1_F_1_) and 96 backcross population (BC_2_F_1_) from the cross between Samgwang and Seolgaeng.

Population	Generation	Number of Plant			*χ*^2^ Value (1:1)
		Total	Homo (A/A)	Hetero (A/G)	
Samgwang × Seolgaeng	BC_1_F_1_	210	110	100	0.48
	BC_2_F_1_	96	41	55	2.04

**Table 4 genes-13-00210-t004:** Analysis of agronomic traits comparing the selected BC_1_F_1_ lines and parents, respectively.

Population	Line	Plant Height (cm)	Culm Length (cm)	Panicle Length (cm)	No. of Tillers
Parent	Samgwang ^§^	95.2 ± 4.6	64.8 ± 6.0	19.7 ± 1.5	9 ± 1
	Seolgaeng	82.0 ± 5.8	51.0 ± 3.3	19.6 ± 1.2	7 ± 2
BC_1_F_1_	SS007	94.2	70.8	19.2	13
	SS017	102.0	79.4	18.8	8
	SS018	94.2	73.8	18.8	8
	SS045	99.4	72.0	22.6	6
	SS083	95.6	63.0	20.0	6
	SS100	109.0	73.6	20.8	6
	SS102	100.2	70.0	21.0	9

^§^ Each test in parent variety was performed in three replicates and values are expressed as average ± standard deviation.

**Table 5 genes-13-00210-t005:** Analysis of agronomic traits comparing the selected BC_2_F_1_ lines and parents, respectively.

Population	Line	Plant Height (cm)	Culm Length (cm)	Panicle Length (cm)	No. of Tillers
Parent	Samgwang	112.3 ± 4.4 ^§^	85.6 ± 2.1	21.9 ± 7.5	10.3 ± 2
	Seolgaeng	116.2 ± 8.5	83.1 ± 3.4	19.1 ± 3.7	10.3 ± 3
BC_2_F_1_	SS50-25	118.0	85.9	16.0	18
	SS50-26	116.8	92.8	18.2	14
	SS50-27	118.3	86.7	17.4	10
	SS50-28	117.5	83.6	18.8	14
	SS29-06	111.5	89.7	19.1	10
	SS29-12	110.3	88.1	19.5	17

^§^ Each test in parent variety was performed in three replicates and values are expressed as average ± standard deviation.

**Table 6 genes-13-00210-t006:** Analysis of viscosity properties, comparing selected BC_2_F_2_ lines and parents.

Population	Viscosity (RVU)										GT ^∫^ (°C)	
	PV ^§^		HPV ^¶^		Breakdown		CPV ^†^		Setback			
	Mean ± SD	*t*	Mean ± SD	*t*	Mean ± SD	*t*	Mean ± SD	*t*	Mean ± SD	*t*	Mean ± SD	*t*
Samgwang	235.2 ± 15.1 ^§^	0.85	139.1 ± 9.8	0.079	120.3 ± 18.3	0.831	219. ± 16.1	5.179 **	−22.6 ± 1.3	−2.478	69.7 ± 1.0	−3.471 *
Seolgaeng	223.1 ± 19.5		138.3 ± 13.9		110.1 ± 11.2		165.7 ± 2.9		−20.4 ± 0.7		72.1 ± 0.7	
SS29-01	205.75		118.58		87.17		196.75		−9.00		72.15	
SS20-02	217.59		126.92		90.67		206.38		−11.21		72.88	
SS29-03	203.79		122.71		81.08		202.09		−1.71		72.40	
SS29-04	235.96		144.29		91.67		227.67		−8.29		71.45	
SS29-05	256.50		144.75		111.75		231.50		−25.00		72.20	
SS29-06	247.25		124.84		122.42		218.46		−28.79		70.33	
SS29-11	246.67		144.16		102.50		228.75		−17.92		72.50	
SS29-12	242.79		129.83		112.96		220.42		−22.37		70.68	
SS29-21	234.75		139.58		95.17		224.00		−10.75		72.10	
SS29-22	229.88		131.96		97.92		213.75		−16.13		71.08	
SS50-20	213.46		139.84		73.63		225.17		11.71		72.13	
SS50-25	241.42		138.42		103.00		221.59		−19.84		72.58	
SS50-26	251.67		130.88		120.79		215.00		−36.67		72.10	
SS50-27	252.25		135.42		116.84		216.92		−35.33		72.10	
SS50-28	238.88		128.38		110.00		217.67		−21.21		72.10	
*F*-value	37.93 ***		27.46 **		78.19 ***		29.81 ***		88.80 ***		19.09 ***	

^§^ PV, peak viscosity; ^¶^ HPV, hot paste viscosity; ^†^ CPV, cool paste viscosity; ^∫^ GT, gelatinization temperature. * *p* < 0.05; ** *p* < 0.01; *** *p* < 0.001. ^§^ Each test in parent variety was performed in three replicates and values are expressed as average ± standard deviation.

**Table 7 genes-13-00210-t007:** Analysis of agronomic traits comparing the selected BC_2_F_2_ lines and parents, respectively.

Population	Line	Plant Height (cm)	Culm Length (cm)	Panicle Length (cm)	No. of Tillers
Parent	Samgwang	109.8 ± 1.65 ^§^	88.7 ± 2.24	18.4 ± 0.16	13 ± 0.80
	Seolgaeng	106.0 ± 0.12	84.4 ± 1.36	19.0 ± 0.55	10 ± 0.98
BC_2_F_2_	SS50-25	108.3 ± 0.62	88.6 ± 0.10	18.5 ± 0.45	11 ± 1.96
	SS50-26	104.8 ± 0.41	85.0 ± 1.25	16.7 ± 0.24	10 ± 0.50
	SS50-27	109.3 ± 0.62	90.8 ± 0.99	17.4 ± 0.50	13 ± 1.41
	SS50-28	110.8 ± 1.74	88.4 ± 1.51	19.5 ± 0.53	13 ± 0.94
	SS29-06	112.6 ± 0.66	94.5 ± 2.02	18.4 ± 0.10	12 ± 1.50
	SS29-12	112.8 ± 0.45	92.8 ± 0.91	18.0 ± 0.20	12 ± 0.82
	*F*-value	5.28 **	7.98 ***	5.31 **	1.23

** *p* < 0.01; *** *p* < 0.001. ^§^ Each test was performed in five replicates and values are expressed as average ± standard deviation.

## Data Availability

Not applicable.

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
