# Peer review of "Marker-Assisted Backcrossing (MABc) to Improve Eating Quality with Thin Seed Coat and Aleurone Layer of Non-Glutinous Japonica Variety in Rice"

_genes, 2022, doi:10.3390/genes13020210_

Round 1

Reviewer 1 Report

Manuscript ID: genes-1561233 titled “Marker-assisted backcrossing (MABc) to improve eating quality with thin seed coat and aleurone layer of non-glutinous japonica variety in rice” focused to improve eating quality of the non-glutinous japonica variety. Rice is a staple food on which depend more than half of the world population for daily source of calories and proteins. Despite this importance, the consumers demand on rice eating quality is increasing raising the need to focus the breeding programs on this trait. From this point of view, the research project described in this manuscript (MS) is relevant. The methodological approach is appropriate and scientifically sound, the experiment design and the software are relevant leading the authors to identify SNP markers associated with improving eating mapped in the SSIIa gene in this study. Based on this findings, the authors suggested that MABc can be used as an additional breeding material for the development of highly processed rice varieties.

The objectives of the study have to be clearly described in the introduction and in the asbract as well. Statistical analyses are needed to support some results recorded in the MS.

For specifics comments see the attached PDF file.

Author Response

Responses to Reviewer 1

We appreciate the comments that the reviewers have given in our manuscript and the constructive criticism the reviewer has given. We have carefully reviewed the comments and have revised the manuscript accordingly. We believe that these changes have clearly improved our manuscript. 

Manuscript ID: genes-1561233 titled “Marker-assisted backcrossing (MABc) to improve eating quality with thin seed coat and aleurone layer of non-glutinous japonica variety in rice” focused to improve eating quality of the non-glutinous japonica variety. Rice is a staple food on which depend more than half of the world population for daily source of calories and proteins. Despite this importance, the consumers demand on rice eating quality is increasing raising the need to focus the breeding programs on this trait. From this point of view, the research project described in this manuscript (MS) is relevant. The methodological approach is appropriate and scientifically sound, the experiment design and the software are relevant leading the authors to identify SNP markers associated with improving eating mapped in the SSIIa gene in this study. Based on this findings, the authors suggested that MABc can be used as an additional breeding material for the development of highly processed rice varieties.

The objectives of the study have to be clearly described in the introduction and in the abstract as well.

--- Thank you for the critical comments. We have described the objectives in Lines 21~23 as follows: In this study, we tried to develop new breeding lines with a thinner seed coat and aleurone layer to provide high eating quality with improved softer chewing characteristics and processability in rice grain.

--- And in Lines 34~35 as follows: These lines will be useful to develop new brown rice varieties with softer chewing characteristics and processability in rice grain.

--- Thank you for the critical comments. We have described the objectives in Lines 83~89 as follows: we tried to develop new breeding lines with a thinner seed coat and aleurone layer to provide high eating quality with improved softer chewing characteristics and processability in rice grain by applying MABc breeding programs to backcross populations between Samgwang and Seolgaeng using KASP markers.

Statistical analyses are needed to support some results recorded in the MS.

--- Thank you for the critical comments. We have included the statistical analyses in Table 4~7 and Figure 7.

Table 4 & 5: BC1F1 and BC2F1 have only one plant. So the statistics analysis cannot be applied.

Figure 7: “Each experiment was performed in three replicates.” Was included in Lines 472-473.

Table 6 & 7: The results of statistics analyses were included in Lines 490~491.

For specifics comments see the attached PDF file.

--- Thank you for the critical comments. We have revised all minor points in the text of the manuscript.

Reviewer 2 Report

Dear Authors,

In reviewed manuscript ‘Marker-assisted backcrossing (MABc) to improve eating quality with thin seed coat and aleurone layer of non-glutinous japonica variety in rice’ Authors made an attempt to investigate possibility of increasing the quality of rice. The topic is interesting and very important.

I recommend English check by native speaker.

Materials and methods:

It would use a general scheme of the experiment to make it clear what, when and how was done.

To all descriptions of methods, please add the number of biological replications.

Info about developing a backcross population should be present in ‘Plant material’ .

Please revise lines 97-103. In first sentence please indicate that measurements were done with ‘near-infrared spectroscopy’. No need to repeat that used method is non-destructive.

Results:

The results are overloaded with information that should be in an introduction, methods, or discussion. Please move them to the right places.

Lines 181-195: this part is introduction not results.

Lines 197-209: please move the descriptions of the tested parameters to 'materials and methods'. There is no need to repeat all the values from table in text (here and everywhere else). Please indicate the significance of differences (here and everywhere else). Does tested parameters have units (here and in every other table/graph)? If yes present them in table. In captions please include number of replicates (here and in every other table/graph). Results are ±SE or SD (here and in every other table/graph)?

Each table/graph must contain units and information about the number of replications.

Lines 217-222: this is introduction or discussion.

Line 222: info about whiteness please move to ‘Material and methods’.

Lines 228-264: there are a lot of text fragments in this section that are either an introduction or a discussion. Do not repeat every value from table in text.

Lines 270-280: remove methodological information from this part.

Lines 293-317: remove methodological information from this part.

Lines 332-346: there are interpretations of the results in this section that should be in discussion.

Lines 350-362: also from here methodological information and interpretations of the results should be moved to other sections of manuscript.

Discussion:

In the discussion, the results should be interpreted in more detail. The parts that are in the results can be useful for this. I would also like to see the summary / conclusions.

With best regards

Author Response

Responses to Reviewer 2

We appreciate the comments that the reviewers have given in our manuscript and the constructive criticism the reviewer has given. We have carefully reviewed the comments and have revised the manuscript accordingly. We believe that these changes have clearly improved our manuscript.  

Reviewer 2:

Dear Authors,

In reviewed manuscript ‘Marker-assisted backcrossing (MABc) to improve eating quality with thin seed coat and aleurone layer of non-glutinous japonica variety in rice’ Authors made an attempt to investigate possibility of increasing the quality of rice. The topic is interesting and very important.

I recommend English check by native speaker.

--- Thank you for the comments. We have checked with a native speaker.

Materials and methods:

It would use a general scheme of the experiment to make it clear what, when and how was done.

--- Thank you for the comments. We have described the points in Materials and Methods in Lines 108~120 as follows: We revised and specified the points about what, when, and where the MABc system using Samgwang and Seolgaeng was performed in the 'MABc Breeding Strategy' section.

To all descriptions of methods, please add the number of biological replications.

--- Thank you for the critical comments. We have revised them in Materials and Methods as follows: We conducted three replicates for each experiment and revised them in 'Materials and Methods', respectively.

Info about developing a backcross population should be present in ‘Plant material’.

--- Thank you for the critical comments. We revised and described the information on backcross population development as shown '2.2. MABc Breeding strategy' in Lines 108~120.

Please revise lines 97-103. In first sentence please indicate that measurements were done with ‘near-infrared spectroscopy’. No need to repeat that used method is non-destructive.

--- Thank you for the critical comments. We deleted the paragraph 'in a non-destructive method'.

Results:

The results are overloaded with information that should be in an introduction, methods, or discussion. Please move them to the right places.

Lines 181-195: this part is introduction not results.

--- Thank you for the critical comments. We have modified the parts in Lines 248~250 as follows: Samgwang is a variety with high yield and high quality which is one of the most popular ones in Korea [33]. Seolgaeng is an excellent rice variety that consumers use as a brown rice.

Lines 197-209: please move the descriptions of the tested parameters to 'materials and methods'. There is no need to repeat all the values from table in text (here and everywhere else). Please indicate the significance of differences (here and everywhere else). Does tested parameters have units (here and in every other table/graph)? If yes present them in table. In captions please include number of replicates (here and in every other table/graph). Results are ±SE or SD (here and in every other table/graph)?

--- Thank you for the critical comments. We described units of analysis values in the 'Materials and Methods' section and annotated below each Table to specify the number of replication and notation of each experiment.

Each table/graph must contain units and information about the number of replications.

--- Thank you for the critical comments. We described units and information on the number of replications in each Table and Figure.

Lines 217-222: this is introduction or discussion.

--- Thank you for the critical comments. We think that the information will be helpful for readers to easily understand by giving nearby.

Line 222: info about whiteness please move to ‘Material and methods’.

--- Thank you for the critical comments. We described also the meaning of whiteness in the 'Materials and Methods' section. And we modified the part as follows: whiteness of grain in rice ---phenotype of rice grain.

Lines 228-264: there are a lot of text fragments in this section that are either an introduction or a discussion. Do not repeat every value from table in text.

--- Thank you for the critical comments. We revised the text repetition for all values by deleting them.

Lines 270-280: remove methodological information from this part.

Lines 293-317: remove methodological information from this part.

Lines 332-346: there are interpretations of the results in this section that should be in discussion.

Lines 350-362: also from here methodological information and interpretations of the results should be moved to other sections of manuscript.

--- Thank you for the critical comments. We believe that the information will be helpful for readers to easily understand the results by giving the explanation nearby.

Discussion:

In the discussion, the results should be interpreted in more detail. The parts that are in the results can be useful for this. I would also like to see the summary / conclusions.

With best regards

--- Thank you for the critical comments. We have modified "Discussion” in Lines 532~609

--- We also include "Conclusion” on Lines 611~623.
